# The neuropeptide substance P regulates aldosterone secretion in human adrenals

Julien Wils [1,2,9], Céline Duparc [1,9], Anne-Françoise Cailleux[3,4], Antoine-Guy Lopez [1,3], Caroline Guiheneuf[3], Isabelle Boutelet[1], Hadrien-Gaël Boyer[1], Christophe Dubessy [1], Saloua Cherifi [1], Bruno Cauliez[5], Françoise Gobet[6], Guillaume Defortescu[7], Jean-François Ménard[8], Estelle Louiset [1,10] & Hervé Lefebvre [1,3,10 ✉]

Aldosterone, produced by the adrenals and under the control of plasma angiotensin and potassium levels, regulates hydromineral homeostasis and blood pressure. Here we report that the neuropeptide substance P (SP) released by intraadrenal nerve fibres, stimulates aldosterone secretion via binding to neurokinin type 1 receptors (NK1R) expressed by aldosterone-producing adrenocortical cells. The action of SP is mediated by the extracellular signal-regulated kinase pathway and involves upregulation of steroidogenic enzymes. We also conducted a prospective proof-of-concept, double blind, placebo-controlled clinical trial aimed to investigate the impact of the NK1R antagonist aprepitant on aldosterone secretion in healthy male volunteers (EudraCT: 2008-003367-40, ClinicalTrial.gov: NCT00977223). Participants received during two 7-day treatment periods aprepitant (125 mg on the 1st day and 80 mg during the following days) or placebo in a random order at a 2-week interval. The primary endpoint was plasma aldosterone levels during posture test. Secondary endpoints included basal aldosterone alterations, plasma aldosterone variation during metoclopramide and hypoglycaemia tests, and basal and stimulated alterations of renin, cortisol and ACTH during the three different stimulatory tests. The safety of the treatment was assessed on the basis of serum transaminase measurements on days 4 and 7. All pre-specified endpoints were achieved. Aprepitant decreases aldosterone production by around 30% but does not influence the aldosterone response to upright posture. These results indicate that the autonomic nervous system exerts a direct stimulatory tone on mineralocorticoid synthesis through SP, and thus plays a role in the maintenance of hydromineral homeostasis. This regulatory mechanism may be involved in aldosterone excess syndromes.

[1] Normandie Univ, UNIROUEN, INSERM, DC2N, 76000 Rouen, France. [2] Rouen University Hospital, Department of Pharmacology, 76000 Rouen, France. [3] Rouen University Hospital, Department of Endocrinology, Diabetes and Metabolic Diseases, 76000 Rouen, France. [4] Rouen University Hospital, Clinical Investigation Centre, INSERM, CIC1404, 76000 Rouen, France. [5] Rouen University Hospital, Department of Biochemistry, 76000 Rouen, France. [6] Rouen University Hospital, Department of Pathology, 76000 Rouen, France. [7] Rouen University Hospital, Department of Urology, 76000 Rouen, France. [8] Rouen University Hospital, Department of Biostatistics, 76000 Rouen, France. [9] These authors contributed equally: Julien Wils, Céline Duparc. [10] These authors jointly supervised this work: Estelle Louiset, Hervé Lefebvre. ✉email: herve.lefebvre@chu-rouen.fr

The human adrenal cortex produces diverse steroid hormones including aldosterone, the end product of the mineralocorticoid synthesis pathway, which is released by the zona glomerulosa, and the glucocorticoid hormone cortisol secreted by the zona fasciculata. Both mineralocorticoid and glucocorticoid secretions are principally under the control of circulating regulatory factors. Whereas cortisol biosynthesis is dependent on the pituitary adrenocorticotrophic hormone (ACTH), the production of aldosterone, which plays a major role in the maintenance of hydromineral homeostasis and blood pressure regulation, is stimulated by the systemic renin angiotensin system (RAS) and plasma potassium[1].

However, it is well established that, in patients receiving prolonged antihypertensive treatment with angiotensin-converting enzyme inhibitors (ACEi) and angiotensin II receptor antagonists (ARA), plasma aldosterone levels tend to re-augment after an initial decrease[2,3]. The mechanism of this phenomenon called "aldosterone breakthrough" remains unexplained. In particular, several clinical studies have shown no association between aldosterone breakthrough and variations of plasma potassium and angiotensin II (Ang II) levels[4,5]. Other regulatory factors unaffected by RAS inhibition, appear thus involved in the maintenance of aldosterone secretion. Especially, it has been shown that a wide variety of conventional neurotransmitters and neuropeptides are able to modulate corticosteroid secretion in vitro[6,7]. These bioactive signals may be released in the adrenal gland by chromaffin cells or the complex network of nerve fibres, comprising both cholinergic and noradrenergic fibres, which has been described in the outer part of the adrenal cortex[8,9]. Interestingly, some of these subcapsular nerve fibres have been shown to be immunoreactive for substance P (SP)[10].

SP belongs to the family of tachykinins, also including neurokinins A and B (NKA and NKB), hemokinin-1 and endokinins. These neuropeptides are involved in the control of pain, emesis, gonadotropic function and participate in the pathogenesis of menopausal hot flushes[11–14]. Although some evidence indicated that SP may stimulate corticosteroidogenesis and exert trophic actions on the rat adrenal cortex[15,16], its role in the regulation of the aldosterone production remains unclear.

Here we show the presence of SP-positive nerve fibres in the human adrenal cortex in the vicinity of aldosterone-producing cells. SP exerts a stimulatory tone on aldosterone secretion through a neurocrine mechanism involving the neurokinin type 1 receptor (NK1R) which is expressed by zona glomerulosa cells. In healthy volunteers, aprepitant, a NK1R antagonist, reduces aldosterone production independently of the RAS.

## Results

**Expression of SP in human adrenal cortex**. We have investigated expression of genes encoding tachykinins (TAC1,3,4) in human adrenals. TAC1 encoding SP and NKA was expressed at high levels whereas TAC3 and TAC4 mRNAs, encoding respectively neurokinin B and endokinins, were unmeasurable or barely detectable (Fig. 1a). Immunohistochemistry showed the presence of SP-positive nerve fibres which were mainly visualised in the zona glomerulosa and, more rarely, in the zona fasciculata (Fig. 1b–d), as previously observed[10]. SP-containing fibres were also visualised in the wall of adrenal arteries in the vicinity of the gland (Supplementary Fig. 1). Although localised in the same intraadrenal nerve trunks, the SP-positive fibres are distinct from adrenergic and cholinergic fibres (Fig. 1e, f). They thus belong to the non-adrenergic non-cholinergic (NANC) nervous system which may represent the third constituent of the autonomic nervous system beside the sympathetic and parasympathetic components[17].

**Expression of tachykinin receptors in human adrenal cortex**. We then investigated whether SP may influence corticosteroidogenesis. The biological actions of tachykinins are known to be mediated by three types of G protein-coupled receptors named NK1, NK2 and NK3 (NK for neurokinin), respectively encoded by the TACR1, TACR2 and TACR3 genes[12]. In addition, alternative splicing of the TACR1 primary transcript generates two isoforms, i.e. the long (TACR1l) and short (TACR1s) variants. The NK1R has a high affinity for SP (EC50 = 1 nM) and weakly binds NKA, while the NK2 receptor (NK2R) is principally activated by NKA but exhibits a low affinity for SP (EC50 = 100 nM)[18,19]. In human adrenals, RT-PCR analyses allowed detection of TACR1l and TACR1s mRNA whereas TACR2 mRNA levels were very low and TACR3 mRNA appeared undetectable (Fig. 2a). Expression and distribution of NK1R in adrenals were investigated by western blot and immunohistochemistry by using three different antibodies recognising the second extracellular loop, the third intracellular region and the cytoplasmic C-terminal tail of the long isoform, respectively (Fig. 2b–d). The long isoform (55 kDa) was clearly detected in all adrenal samples with antibodies specific to the C-terminal region of the protein (Fig. 2b–d, Supplementary Fig. 2). The short isoform (<40 kDa) was also observed in some specimens with the antibodies against the third intracellular region of the receptor. In addition, the three antibodies revealed the presence of a protein of higher molecular weight (70 kDa) which likely corresponds to the glycosylated, phosphorylated and/or ubiquitinated forms of the NK1R, as previously reported[20,21]. In some extracts, antibodies directed to the C-terminal region of the receptor allowed detection of an additional high molecular band (115 kDa) which has been identified as an ubiquitinated variety of the protein[20]. Intense NK1R immunoreactivity was detected in the adrenal cortex and in sympathetic ganglia and arteriole walls located at the periphery of the gland (Fig. 2b–d; Supplementary Fig. 3). NK1R immunoreactivity was principally observed in the zona glomerulosa whereas much weaker labelling was detected in the zona fasciculata. In addition, NK1R-positive adrenocortical cells were located close to nerve fibres containing SP (Fig. 2e). Interestingly, aldosterone-producing cells, which thus express aldosterone synthase encoded by CYP11B2, were labelled by NK1R antibodies (Fig. 2f, g). The NK1R is also detected in zona glomerulosa cells negative for CYP11B2, which have formerly been shown to express CYP11B1, a key enzyme for cortisol production[22]. These immunohistochemical studies strongly suggested that tachykinins may modulate corticosteroidogenesis through activation of the NK1R.

**SP stimulates aldosterone production in vitro**. We have thus investigated the action of SP on corticosteroid production from human adrenocortical cells in primary culture. SP dose-dependently stimulated aldosterone secretion with similar efficacy but higher potency than NKA (Fig. 3a; EC50 = 1.3 ± 0.3 nM versus 27.3 ± 0.3 nM; p < 0.0001; Mann–Whitney test). SP also stimulated cortisol release but with lower potency (Fig. 3b; EC50 = 76.7 ± 0.3 nM; p < 0.0001; Supplementary Table 1). The effect of SP on aldosterone secretion was blunted by the NK1R antagonist aprepitant (Fig. 3c). By contrast, GR159897, a NK2R antagonist, did not affect the aldosterone response to SP ($10^{-6}$ M; mean ± SEM: 185 ± 23%, CI 95%: 110–260 versus 227 ± 21%, CI%: 158–295; p = 0.34; n = 4; Fig. 3d). Globally, these data thus show that, in the human adrenal, SP enhances steroidogenesis through activation of NK1R, its biological action being more potent to stimulate aldosterone than cortisol secretion.

**Mechanisms of action of SP on human adrenocortical cells**. The intracellular calcium signalling pathway is a major regulator of

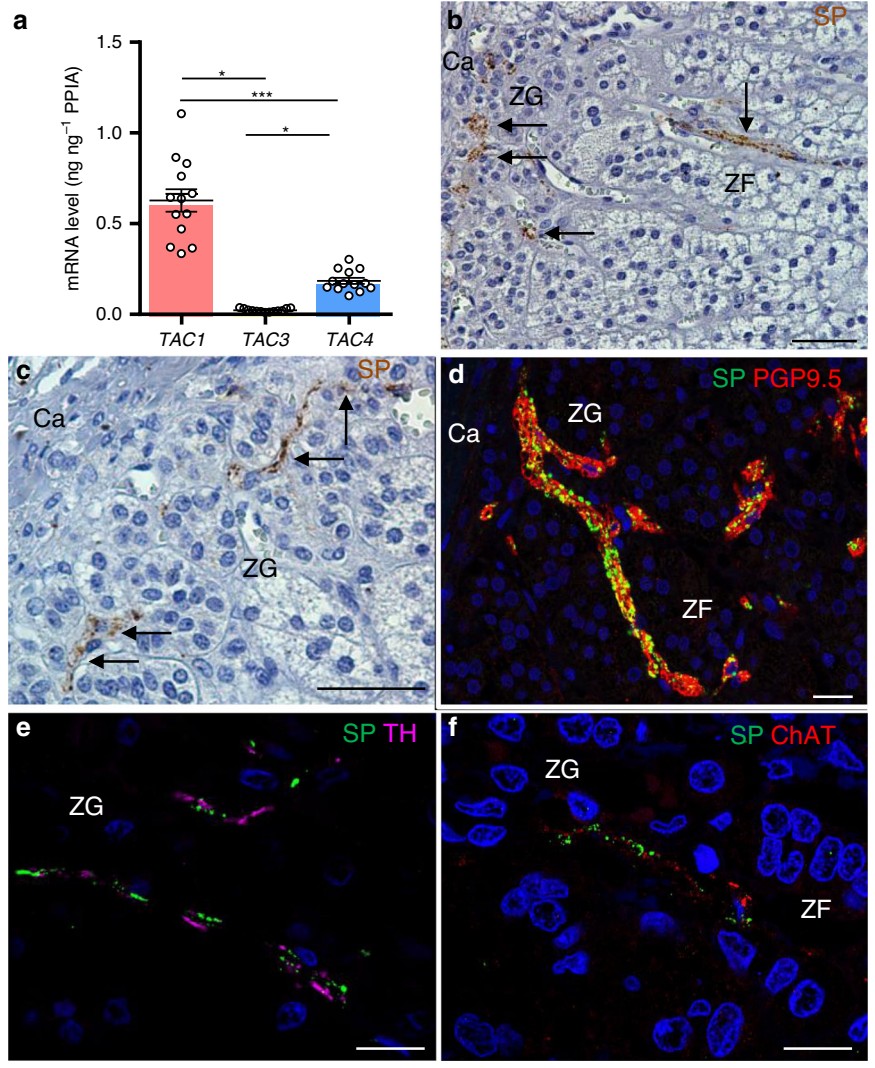

**Fig. 1 Expression of substance P (SP) in the human adrenal gland. a** Quantitative RT-QPCR analysis of *TAC1*, *TAC3* and *TAC4* mRNAs ($n = 13$ independent adrenals). Each dot indicates one adrenal. Data are presented as mean ± SEM. CI 95%: 0.49–0.76, 0.017–0.028, 0.15–0.22 for *TAC1*, *TAC3* and *TAC4*, respectively. Statistical analysis was performed by Kruskal–Wallis test and Dunn's post-test after one-way ANOVA. Significance denoted by *** $p < 0.001$, *$p < 0.05$. **b**, **c** Distribution of SP immunoreactivity in the adrenal cortex. Arrows designate nerve fibres. (Ca capsule, ZG zona glomerulosa, ZF zona fasciculata; scale bars= 50 µm; microphotographs representative of $n = 18$ independent adrenals). **d–f** Double immunofluorescence detection of SP and the nerve fibre marker PGP9.5 (**d**), tyrosine hydroxylase (TH; **e**), a marker of adrenergic nerve fibres, or choline acetyltransferase (ChAT; **f**), a marker of cholinergic nerve fibres. (scale bars = 20 µm; microphotographs representative of $n = 4$, 14 and 4 independent adrenals in **d–f**, respectively).

aldosterone synthesis, both in physiological and pathophysiological conditions[1]. As a matter of fact, the stimulatory effect of angiotensin II (AngII) on aldosterone secretion is principally mediated by an increase in cytosolic calcium concentration $[Ca^{2+}]_c$. By contrast, high concentration of SP weakly increased $[Ca^{2+}]_c$ in cultured adrenocortical cells, corresponding only to 18 ± 3% of the AngII-induced $Ca^{2+}$ response (Fig. 4a, b). The high potency of SP (EC50 = 5.9 ± 0.4 nM) to increase $[Ca^{2+}]_c$ is consistent with activation of NK1R[19].

NK1Rs are also known to stimulate the ERK/MAP kinase pathway[23] which is involved in the aldosterone response to AngII and other factors such as adipocyte secretory products[1,24]. Consistently, our data show that SP dose- and time-dependently stimulates ERK phosphorylation in cultured adrenocortical cells (EC50 = 1.1 ± 0.1 nM at 5 min; Fig. 5a–d; Supplementary Fig. 4). The high potency of SP to activate ERK phosphorylation is indicative of the involvement of NK1R in the biological response. In addition, SP-evoked aldosterone secretion

is blocked by the ERK inhibitor PD0325901 (145 ± 18%, CI 95%: 86–204 versus 85 ± 11%, CI 95%: 75–124; $n = 4$; $p = 0.02$; Fig. 5e). We have then investigated the impact of SP on the expression of the genes involved in aldosterone synthesis in comparison with AngII (Fig. 5f; Supplementary Fig. 5). SP had no significant effect on the levels of *STAR* and *CYP11B2* transcripts but increased the rates of *HSD3B2* and *CYP21A2* mRNAs, encoding the steroidogenic enzymes 3βHSD2 and 21-hydroxylase. Enhanced expression of 3βHSD enzymes has been shown to result in aldosterone overproduction in mice[25] through an increase in substrates for aldosterone synthase. It appears thus as a reliable mechanism to mediate the impact of SP on zona glomerulosa steroidogenesis.

**Aprepitant reduces aldosterone production in vivo.** Complementary in vivo studies were necessary to determine whether SP actually exerts a physiological role in the regulation of aldosterone secretion. NK1R antagonists, like aprepitant, are now

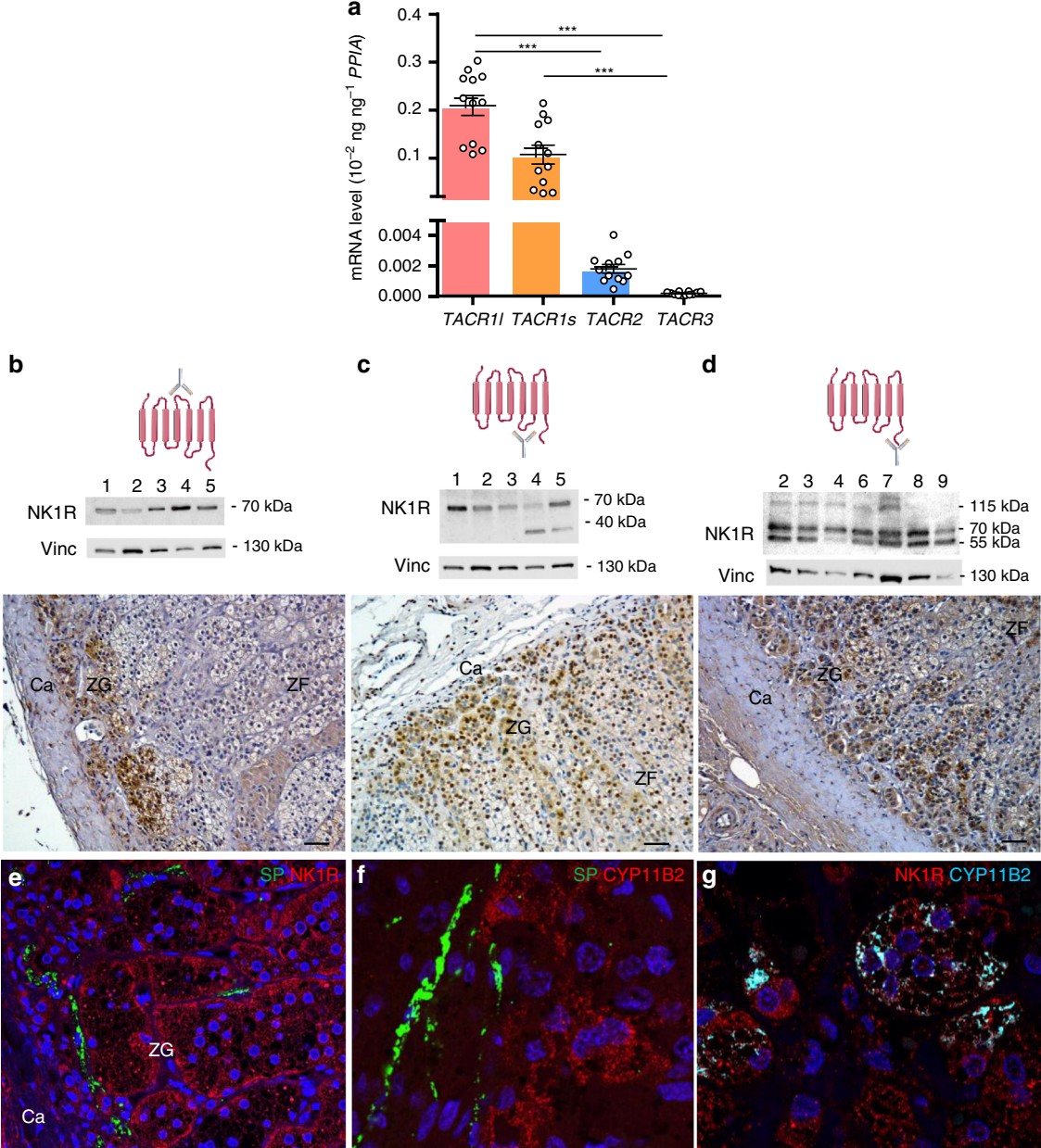

**Fig. 2 Expression of tachykinin receptors in the human adrenal gland. a** Quantitative RT-PCR analysis of *TACR1l*, *TACR1s*, *TACR2* and *TACR3* mRNAs ($n = 12$ independent adrenals). Each dot indicates one adrenal. Data are presented as mean ± SEM. CI 95%: 0.16–0.25, 0.06–0.15, 0.001–0.002 and 0.0001–0.0002 for *TACR1l*, *TACR1s*, *TACR2* and *TACR3*, respectively. Statistical analysis was determined by Kruskal–Wallis test and Dunn's post-test after one-way ANOVA. Significance denoted by ***$p < 0.001$. **b**–**d** Presence of neurokinin type 1 receptor (NK1R) visualised by western blot and immunohistochemistry by using three antibodies recognising different epitopes (diagrams on the higher panels). Representative western blots showing bands of different molecular weights detected in adrenal extracts (middle panels; $n = 5$ adrenals in **b** and **c**; $n = 7$ adrenals in **d**). Vinculin (Vinc) was used as a loading control. Distribution of NK1R immunostaining in the cortex (lower panels; scale bars = 50 µm; Ca capsule, ZG zona glomerulosa, ZF zona fasciculata, microphotographs representative of $n = 10$, 7 and 10 adrenals in **b**–**d**, respectively). Schemas created using medical diagrams available from https://smart.servier.com under a CC BY 3.0 licence (https://creativecommons.org/licenses/by/3.0/). **e**–**g** Immunofluorescence detection of SP (**e**, **f**) and NK1R (**e**, **g**) or CYP11B2 (aldosterone synthase; **f**, **g**) in the ZG (scale bars = 20 µm; microphotographs representative of $n = 4$ adrenals).

commonly used in the treatment of chemotherapy-induced nausea and vomiting[26]. We have conducted a prospective placebo-controlled double-blind study aimed at investigating the impact of aprepitant per os on corticosteroid levels in 20 healthy volunteers (Supplementary Table 2; Supplementary Fig. 6). Aprepitant had no effect on urinary cortisol level ($p = 0.18$) nor on plasma cortisol ($p = 0.89$) and ACTH ($p = 0.43$) levels in basal condition or in response to hypoglycaemia stress (Fig. 6a–c;

Supplementary Discussion; Supplementary Figs. 7 and 8). It seems thus that the previously reported stimulatory effect of SP on ACTH production[27] does not involve the NK1R. Conversely, aprepitant reduced by 28% mean global aldosterone production assessed by measurement of daily urinary aldosterone excretion ($43 \pm 4$ versus $31 \pm 2$ nmol for 24 h; $p = 0.0009$; Fig. 6d). The NK1R antagonist had no influence on plasma aldosterone levels in upright position ($p = 0.48$) but significantly reduced

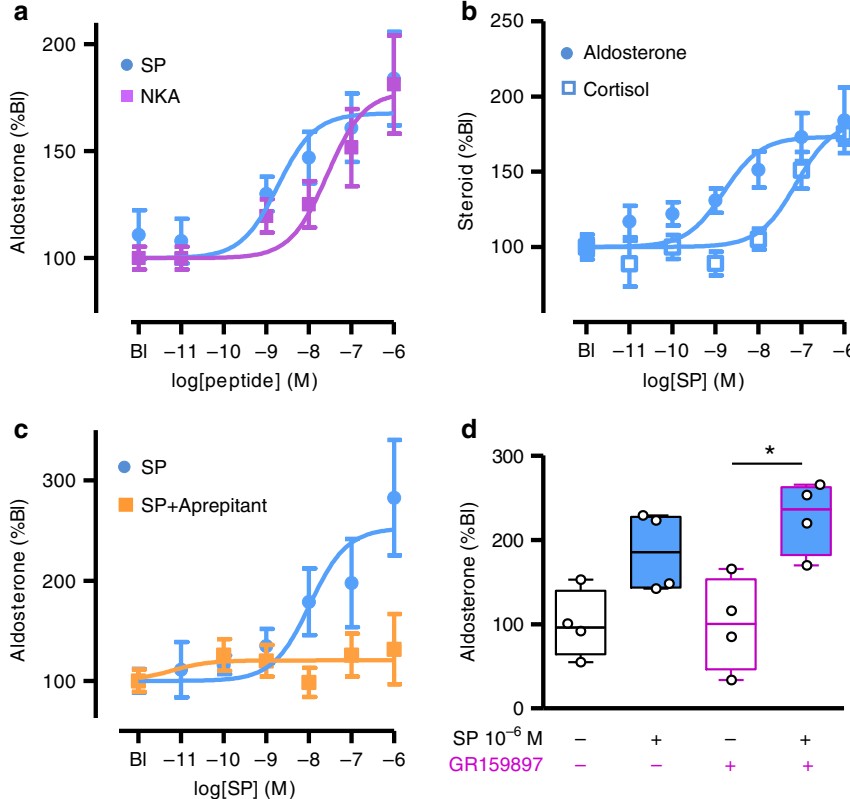

**Fig. 3 Effect of tachykinins on steroidogenesis in cultured adrenocortical cells. a** Effects of SP ($n = 23$ independent cultures performed in quadruplicate) and neurokinin A (NKA; $n = 5$ independent cultures performed in quadruplicate) on aldosterone production. **b** Effect of SP on aldosterone and cortisol secretion ($n = 5$ independent cultures performed in quadruplicate). **c** Effect of SP on aldosterone secretion in the absence or presence of the NK1R antagonist aprepitant ($10^{-9}$ M; $n = 7$ independent cultures performed in quadruplicate). Dose-response curves were analysed by two-way ANOVA. Aprepitant inhibited the aldosterone response ($F = 10.14$; DFn=1; DFd=72; $p = 0.002$). F distribution, DFn degrees of freedom for groups, DFd degrees of freedom for samples. Data are presented as mean ± SEM in panels **a**–**c**. **d** Effect of the NK2R antagonist GR159897 ($10^{-9}$ M) on the aldosterone response to SP ($10^{-6}$ M). Data are presented as median ± interquartile range (IQR), minima and maxima ($n = 4$ cultures). They were analysed by two-tailed Mann–Whitney test. *$p = 0.028$. In all culture experiments, steroid secretion was normalised to basal level (Bl).

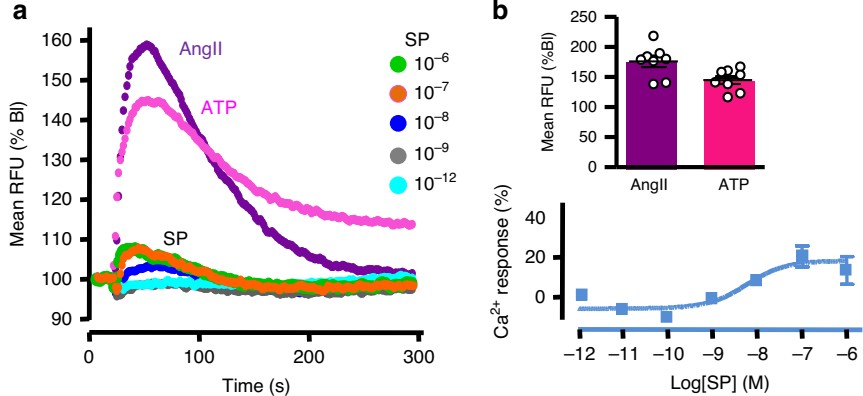

**Fig. 4 Effect of substance P on the calcium signalling pathway in cultured adrenocortical cells. a** Effect of SP (from $10^{-12}$ to $10^{-6}$ M), angiotensin II (Ang II; $10^{-6}$ M) and ATP ($10^{-5}$ M) on cytosolic calcium concentrations in cell pools (RFU relative fluorescence unit). Data are normalised to basal level (Bl) and expressed as mean of three culture dishes (triplicate). Recordings are representative of $n = 3$ independent cultures. **b** Maximum amplitude of Ang II- and ATP-triggered calcium responses (upper panel). Data are presented as mean ± SEM of eight cultures dished examined over three independent experiments. CI 95% were 154–197 and 129–160 for Ang II and ATP, respectively. Lower panel: dose-response curve of SP-induced maximum calcium responses normalised to those of Ang II illustrated in panel **a**.

plasma aldosterone level in recumbency ($322 \pm 31$ versus $239 \pm 17$ pmol L$^{-1}$; $p = 0.04$; Fig. 6e, f). In addition, aprepitant had no influence on plasma renin ($p = 0.12$), potassium levels ($p = 0.12$) and blood pressure (Fig. 6g; Supplementary Figs. 9 and 10).

Surprisingly, aprepitant potentiated the aldosterone secretion induced by metoclopramide, a serotonin receptor type 4 receptor agonist ($p = 0.002$; two-way ANOVA, Supplementary Discussion; Supplementary Fig. 11).

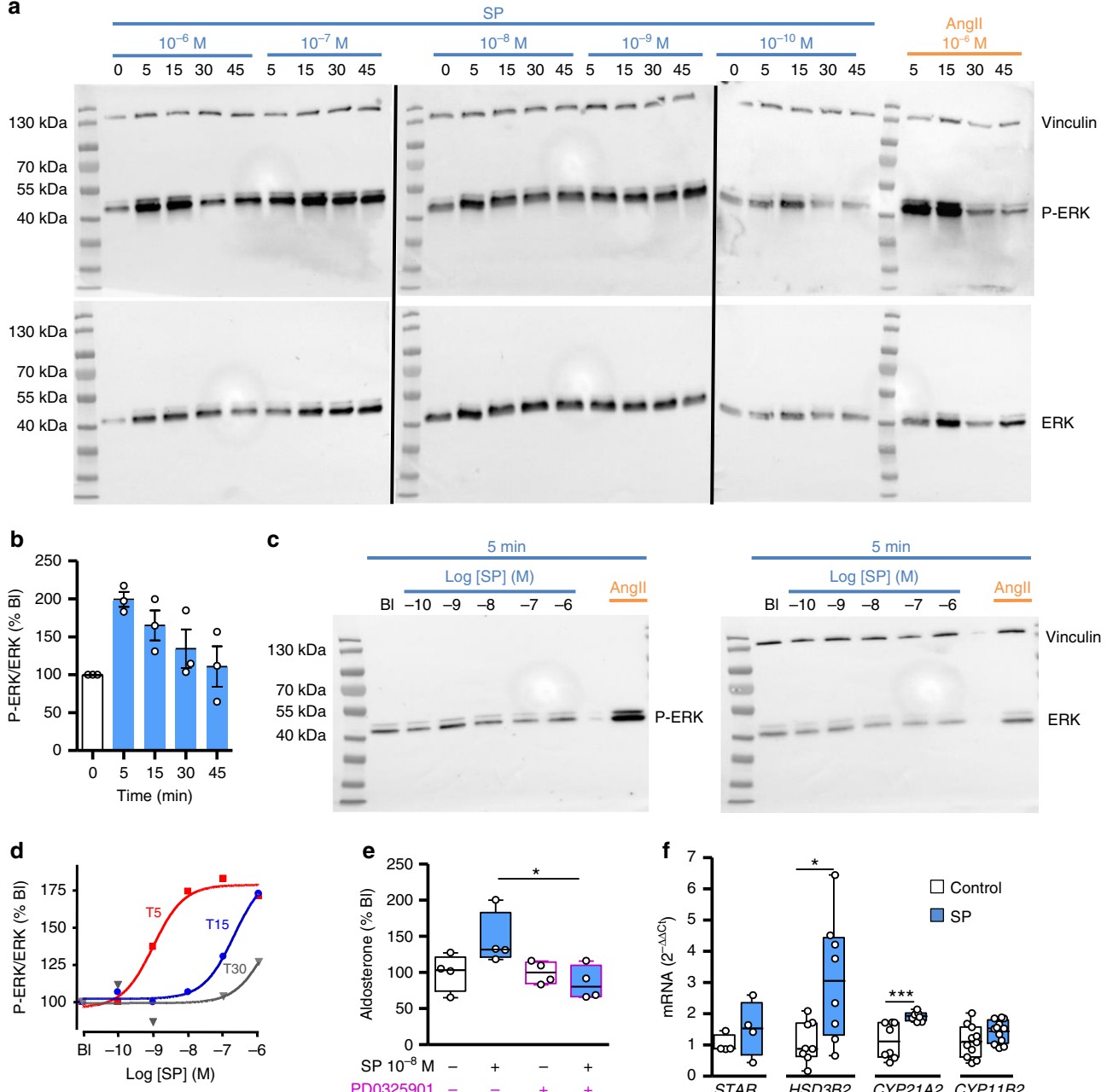

**Fig. 5 Coupling of tachykinin receptors to ERK signalling pathway in cultured adrenocortical cells. a** Representative western blots showing the kinetics of the phospho-ERK (upper panels) and ERK (lower panel) levels in response to SP (from $10^{-6}$ to $10^{-10}$ M) and Ang II ($10^{-6}$ M) after incubation of cultured cells for 5, 15, 30 and 45 min (three blots processed in parallel showing samples derived from the same culture experiment). Immunolabellings for phospho-ERK and ERK were successively performed on the same membranes. Ang II and vinculin were used as positive and loading controls, respectively. **b** Time course of ERK phosphorylation normalised to ERK in response to SP ($10^{-7}$ M). Data are normalised to basal level (0 min) of each repetitive experiment. Data are presented as mean ± SEM of three independent cultures. **c** Representative western blots showing the effect of increasing doses of SP administrated for 5 min on phospho-ERK (left panel) and ERK (right panel). **d** Dose-response curves of SP-induced ERK phosphorylation normalised to ERK for 5, 15 and 30 min of incubation (T5, T15 and T30), illustrated in **c** and Supplementary Fig. 4. Data are expressed as percentage basal level (Bl). **e** Effect of the ERK pathway inhibitor PD0325901 ($10^{-8}$ M) on SP-induced aldosterone secretion. Data are expressed as median ± IQR, minima and maxima ($n = 4$ cultures). They were analysed by two-tailed Mann–Whitney test ($p = 0.028$ for SP versus SP + PD0325901). **f** Effect of SP ($10^{-6}$M, 24 h) on the expression levels of genes encoding cholesterol transporter and steroidogenic enzymes. mRNA expression levels were normalised to PPIA. Data are expressed as median ± IQR, minima and maxima ($n = 4$, 8, 8 and 11 independent cultures for *STAR*, *HSD3B2*, *CYP21A2* and *CYP11B2*, respectively). They were analysed by two-tailed Mann–Whitney test ($p = 0.48$ for *STAR*, $p = 0.02$ for *HSD3B2*, $p = 0.0002$ for *CYP21A2* and p = 0.14 for *CYP11B2*. *$p < 0.05$; ***$p < 0.001$.

## Discussion

The present study indicates that SP exerts a stimulatory tone on aldosterone secretion in man. This effect likely involves SP released by intraadrenal nerve fibres in the immediate vicinity of steroidogenic cells since the plasma levels of the peptide do not normally exceed 0.01 nM[28], a concentration which has no action on aldosterone production in our adrenocortical cell culture experiments. Consequently, it can be proposed that the

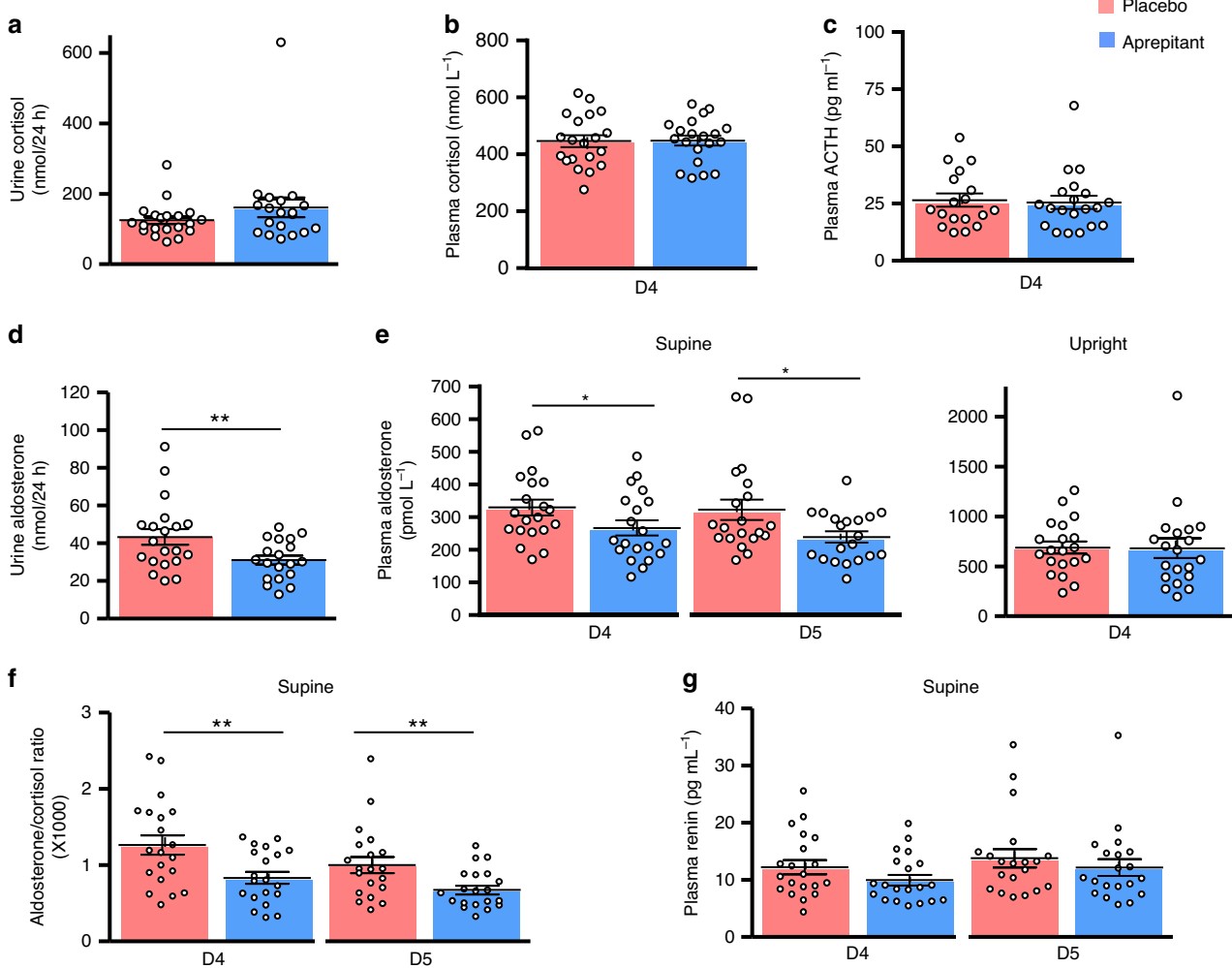

**Fig. 6 Effect of oral administration of the NK1R antagonist aprepitant versus placebo on corticosteroid concentrations in healthy volunteers. a–c** Effect of placebo or aprepitant on 24-h urinary cortisol excretion (CI 95%: 102–147 versus 103–219; $p = 0.18$), basal (8:00 a.m.) plasma cortisol (CI 95%: 403–489 versus 411–485; $p = 0.89$) and ACTH levels (CI 95%: 20–32 versus 19–31; $p = 0.43$) at day 4 (D4; 4th day of treatment). **d, e** Effect of placebo or aprepitant on 24-h urine aldosterone (CI 95%: 34–52 versus 26–36; $p = 0.0009$), supine plasma aldosterone levels at day 4 (CI 95%: 278–380 versus 218–315; $p = 0.04$) and day 5 (D5; CI 95%: 256–388 versus 203–274: $p = 0.02$), and plasma aldosterone concentration in upright position at D4 (CI 95%: 562–814 versus 475–888; $p = 0.48$). **f, g** Effect of placebo or aprepitant on supine plasma aldosterone to cortisol ratio (CI 95%: 0.99–1.53 versus 0.67–0.99; $p = 0.004$ at D4 and CI 95%: 077–1.22 versus 0.55–0.79; $p = 0.002$ at D5) and plasma renin levels (CI 95%: 9.6–14.8 versus 8.0–11.9 at D4; $p = 0.12$ and CI 95%: 10.4–17.2 versus 9.1–15.2; $p = 0.28$ at D5). Each dot indicates one healthy volunteer ($n = 20$ healthy volunteers). Data are presented as mean ± SEM. Data were analysed by non-parametric Wilcoxon/mid rank matched pairs test. *$P < 0.05$, **$P < 0.01$.

autonomous nervous system directly controls aldosterone production and thus participates in the regulation of hydromineral homeostasis. From an evolutionary perspective, it is interesting to notice that the control of aldosterone production by tachykinins exists in amphibians and rodents[15,29], its conservation among species being highly indicative of its physiological importance. The molecular, immunohistological and pharmacological studies presented herein, especially the inhibitory action of aprepitant on SP-induced aldosterone production, indicate that the effect of SP on adrenocortical cells is mediated by NK1R. At variance with Ang II which principally stimulates mineralocorticoid production through mobilisation of the calcium signalling pathway, SP has a minor influence on cytosolic calcium concentration. Conversely, the observations that SP activates ERK phosphorylation and that the stimulatory action of SP on aldosterone secretion is blunted by ERK inhibitor suggest that SP controls aldosterone synthesis via activation of the intraadrenal ERK pathway. However, coupling of the NK1R to the ERK pathway in adrenocortical cells needs to be further explored by additional experiments, notably

by investigating the impact of NK1R antagonists on SP-induced ERK phosphorylation.

The observation that antagonising in vivo the adrenal NK1R leads to a decrease in recumbent plasma aldosterone levels suggests that the autonomous nervous system is involved in the maintenance of basal mineralocorticoid production via intraadrenal release of SP. This original mechanism of regulation seems to be complementary to the systemic RAS whose action is predominant in situations necessitating strong stimulation of aldosterone production, like adaptation to the upright position and salt deprivation. The occurrence of SP-positive nerve fibres together with expression of the NK1R in adrenal arteriole walls indicates that, in addition to inhibition of SP action on adrenocortical cells, aprepitant may also decrease aldosterone production through blockade of the vasorelaxant effect of SP and subsequent reduction of the adrenal blood flow which is a major activator of corticosteroidogenesis[30]. The mechanisms of action of SP to control aldosterone production are schematized in Fig. 7.

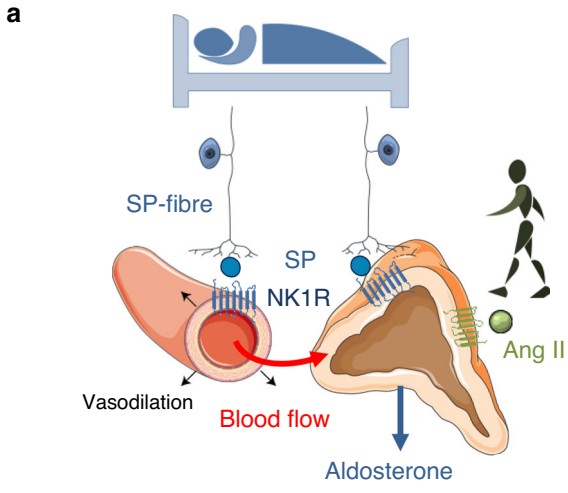

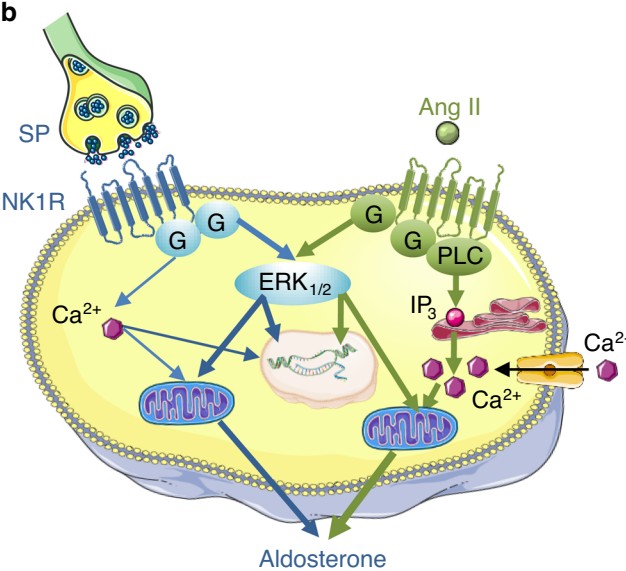

**Fig. 7 Schematic representation of the putative action of substance P (SP) on aldosterone production by the human adrenal gland. a** SP may stimulate aldosterone secretion through both an indirect vascular effect leading to an increase in adrenal blood flow and a direct action on adrenocortical cells, the two mechanisms being mediated by the NK1 receptor (NK1R). The action of SP seems to be complementary to that of Ang II which is involved in the aldosterone response to upright posture while SP may mainly control basal aldosterone production. **b** At the cellular level, binding of SP to the NK1R expressed by zona glomerulosa cells induces an activation of the ERK pathway but has a minor effect on the calcium signalling pathway in contrast to angiotensin II (Ang II) which strongly activates the two transduction mechanisms. Schemas created using medical diagrams available from https://smart.servier.com under a CC BY 3.0 licence (https://creativecommons.org/licenses/by/3.0/).

The lack of increase in plasma renin and potassium levels during aprepitant treatment may appear inconsistent with the decrease in plasma aldosterone concentration. It is likely that the duration of aprepitant treatment was too short to allow elevation of renin and potassium concentrations in response to partial inhibition of aldosterone release. It is also possible that the absence of compensatory increase in renin levels results from the fact that aprepitant diminished aldosterone secretion only in recumbency, a position which is known to physiologically reduce renin production. In addition, our in vivo proof-of-concept study revealed that urinary cortisol levels tend to increase under aprepitant administration, while remaining in the normal range. This unexpected effect of the NK1R antagonist probably results from an impact of the drug at the hypothalamo-pituitary level since antagonising the stimulatory action of SP at the adrenal level should logically induce a decrease in glucocorticoid production. We cannot exclude that aprepitant-evoked cortisol production may become significant during prolonged treatments. It will thus be necessary to perform additional studies to evaluate the impact of long-term administration of aprepitant on glucocorticoid synthesis in addition to renin and aldosterone levels.

From a pathophysiological point of view, activation of the SPergic control of the mineralocorticoid function may be involved in idiopathic aldosteronism related to obesity and sleep apnea syndrome (SAS)[31,32]. Indeed, aldosterone excess appears to be at least partly independent of the RAS in these two conditions which are known to be associated with an increase in activity of the autonomous nervous system[33,34]. It is also relevant to speculate that the tachykininergic innervation of the adrenal cortex may be implicated in essential forms of low renin hypertension[35] and aldosterone breakthrough. NK1R antagonists like aprepitant may thus represent a valuable therapeutic option to reduce obesity- and SAS-associated hyperaldosteronism, manage low renin hypertension and limit aldosterone breakthrough. In fact, it is now well demonstrated that aldosterone exerts profibrotic and inflammation actions in the cardiovascular system and kidney, and counteracting its effects has proved its efficacy to reduce the progression of heart failure and chronic nephropathy[36,37]. Especially, NK1R antagonists may constitute an alternative to both mineralocorticoid receptor antagonists whose utilisation is hampered by their antiandrogenic properties[38,39], and aldosterone synthase inhibitors whose administration leads to accumulation of steroid precursors with mineralocorticoid activity[40], a phenomenon which may reduce their clinical benefits. Our findings open therefore a vast field of clinical investigation aimed at evaluating the action of NK1R antagonists on aldosterone production and organ protection in various cardiovascular, renal and metabolic diseases, potentially enlarging their clinical use from antiemesis to cardiorenal prevention.

## Methods

**Tissues**. Adrenal glands were obtained from patients undergoing expanded nephrectomy for kidney cancer and brain-dead organ donors. Adrenals were collected at surgery and immediately dissected by the pathologist. Adrenal explants were immersed in culture medium until cell dissociation, frozen for RT-PCR and western blot analyses, or fixed in formalin for histological study. The protocol of collection of the tissues and the experimental procedures were approved by the French National Biomedicine Agency (Agence de Biomédecine) and the regional ethics committee. Written informed consents were obtained from all subjects or patients' closest relatives.

**Cell culture**. After dissociation with collagenase type 1A and desoxyribonuclease 1 type 4 (Sigma–Aldrich, Saint-Quentin-Fallavier, France), adrenocortical cells were cultured at a density of $10^6$ cells ml$^{-1}$ in culture medium (50% DMEM, 50% Ham–F12; ThermoFisher Scientific, Illkirch, France) supplemented with 1% antibiotic–antimycotic solution, 1% insulin–transferrin–selenium solution (ThermoFisher) and 5% fetal calf serum (Sigma–Aldrich) at 37 °C in a 5% $CO_2$–95% air atmosphere with 100% relative humidity. The culture medium was changed 24 h after plating for reducing fetal calf serum to 1%.

**Real-time RT-PCR**. Total RNA from adrenal glands or cultured adrenocortical cells was extracted using Tri Reagent (Sigma–Aldrich) and purified on Nucleospin RNAII (Macherey–Nagel, Hoerdt, France). Human adrenal, spinal cord, small intestine, placenta polyA mRNAs (Clontech, Ozyme, Montigny-le-Bretonneux, France), as well as RNA extracted from LAD2 and Caco2 cell lines [given by Dr. D Metcalfe (National Institute of Allergy and Infectious Disease, National Institutes of Health, Bethesda, MD) and Dr Moïse Coeffier (Normandie Univ, UNIROUEN, INSERM, Rouen, France), respectively] were also used as control specimen. Purified RNA and polyA mRNAs were converted into cDNA by using ImProm-II RT System (Promega). Real-time PCR amplifications were performed using SYBR Green I Master Mix (Applied Biosystem, Courtaboeuf, France) in a QuantStudio 3 System (ThermoFisher) with specific primers (Supplementary Table 3). Samples

were analysed in duplicates. Quantification of cDNAs was normalised to *PPIA* (cyclophilin) by using standard curves established by using dilution series of cDNAs derived from polyA mRNAs or *TACR1* human qPCR Template (HK210362, Origene). For cell studies, adrenocortical cells were cultured in six-well plates ($2 \times 10^6$ cells/well) and incubated with fresh DMEM (Control) or DMEM with SP (Sigma-Aldrich). Incubation experiments were conducted in triplicate at 37 °C for 24 h. Gene expression was normalised to *PPIA* by using the average ΔCt value of the basal culture condition (ΔΔCt). The final fold expression changes were calculated using the equation $2^{-\Delta\Delta Ct}$.

**Immunohistology.** Immunohistology was performed on formalin fixed and deparaffinized tissue sections. Sections were heated at 95 °C for 20 min in 10 mM citrate buffer (pH 6) and/or Tris EDTA (pH 9) for antigen retrieval. For immunohistochemical and immunohistofluorescence experiments, tissue sections were treated with peroxidase blocking reagent (Dako Corporation, Les Ulis, France) and normal donkey serum, respectively. Sections were then successively incubated with primary antibodies (Supplementary Table 4) and anti-immunoglobulin coupled to peroxidase or fluorescent-conjugated anti-immunoglobulin antibodies. For immunohistochemical studies, immunoreactivities were revealed with diamino-benzidine (Dako Corporation). The tissue sections were counterstained with hematoxylin and examined on an Eclipse E600D microscope (Nikon). For immunohistofluorescence experiments, nuclei were stained with DAPI (1 pg/ml; Sigma-Aldrich) and fluorescence was examined on a TCS SP8 MP confocal microscope (Leica Corp., Heidelberg, Germany). All images were obtained on PRIMACEN, the Cell Imaging Platform of Normandie, University of Rouen Normandie.

**Hormone secretion.** For hormone secretion studies, adrenocortical cells were cultured in 24-well plates ($5 \times 10^5$ cells/well). Cultured cells were incubated with fresh DMEM (Basal conditions, Bl) or DMEM with different concentrations of SP (Sigma-Aldrich) or neurokinin A (Enzo Life Sciences, Lyon, France). SP was applied in the absence or presence of neurokinin receptor antagonists, aprepitant ($10^{-9}$ M; Selleck Chemicals; Houston, USA) and GR159897 ($10^{-9}$ M; Tocris Bioscience; Bio-Techne Europe, Lille, France) or the MEK inhibitor PD0325901 ($10^{-8}$ M; Sigma-Aldrich). Incubation experiments were conducted in quadruplicate at 37 °C for 24 h. Aldosterone and cortisol concentrations in culture supernatants were measured by means of radioimmunoassay procedure (RIA) using specific antibodies and tritiated steroid hormones (Perkin Elmer, Villebon-sur-Yvette, France) and Tri-Carb 2600TR liquid scintillation counter (Perkin Elmer, Villebon-sur-Yvette, France). RIA sensitivities were 80 pg mL$^{-1}$ and 150 pg mL$^{-1}$ for aldosterone and cortisol, respectively. Cross-reactivity of aldosterone antibodies with corticosterone, cortisol, testosterone and Δ4-androstenedione were <0.001%. Cross-reactivity of cortisol antibodies was <0.001% for aldosterone, <0.004% for progesterone and 17-hydro-xyprogesterone, <0.01% for desoxycorticosterone, <0.3% for cortisosterone and <5% for desoxycortisol. Hormone secretion was normalised to basal level (Bl).

**Cytosolic calcium concentration.** Intracellular Ca$^{2+}$ concentrations were measured on adrenocortical cells cultured in 96-well plates ($2 \times 10^5$ cells/well) by using a scanning fluorometer Flexstation III (Molecular Devices, Sunnyvale, CA) and the SoftMax Pro software on PRIMACEN. Before recording, cultured cells were loaded with FLIPR Calcium 5 Assay Kit (Molecular Devices) during 1 h at 37 °C. The wavelengths of excitation and emission were 485 nm and 525 nm, respectively. SP was added at final concentrations ranging from $10^{-12}$ to $10^{-6}$ M. Angiotensin II (AngII; Sigma-Aldrich) and adenosine triphosphate (ATP; Sigma-Aldrich) were used as positive controls. Incubation experiments were conducted in triplicate. Calcium signals are represented as mean of fluorescence intensities simultaneously recorded in three distinct wells and normalised to basal level. Peaks of SP calcium responses were normalised to maximum AngII-evoked calcium signal.

**Western blot.** Western blot experiments were performed on proteins extracted from adrenal glands and cultured adrenocortical cells incubated at 37 °C for 5, 15, 30 or 45 min with fresh DMEM (control experiments) or DMEM supplemented with different concentrations of SP. Angiotensin II ($10^{-6}$ M) was used as positive control. Incubation experiments were conducted in duplicate. After incubation periods, cultured cells were rinsed once in PBS at 37 °C. Proteins were extracted from adrenal tissues and adrenocortical cell cultures by using radio-immunoprecipitation assay (RIPA) buffer (Thermo Scientific) supplemented with proteinase inhibitors and phosphatase inhibitors mixture 1 and 2 (Sigma-Aldrich). Protein concentrations were determined using Bradford protein assay (Bio-Rad; Marnes-la-Coquette, France). Proteins were diluted in Laemmli buffer (Bio-Rad), separated on mini-protean TGX stain free gels 12% (Bio-Rad) and transferred onto nitrocellulose membrane by using a Transblot turbo blotting system (Bio-Rad). Membranes were successively incubated with 5% non-fat dry milk in TBS-Tween, primary antibodies (overnight at 4 °C), and HRP-conjugated secondary antibodies (Supplementary Table 5). NK1R/vinculin and ERK/phospho-ERK immunolabel-lings were successively performed on the same membranes after an intermediate incubation period of 15 min with Restore western blot stripping buffer (Thermo Scientific). In all experiments, vinculin immunolabeling was used as a loading control. Immunoreactivities were revealed by chemiluminescence with clarity

Western ECL substrate and visualised on ChemiDoc Imaging Systems (Bio-Rad). Signals were quantified with Image lab (Bio-Rad). Expression was normalised to ERK and basal level.

**Clinical trial**

*Protocol design.* A prospective proof-of-concept, double blind, cross-over and placebo-controlled study was conducted in healthy volunteers (Supplementary note 1; APHOS study: Protocol no 2007/049/HP; EudraCT: 2008-003367-40, registration date: 20/05/2008; ClinicalTrial.gov: NCT00977223, registration date: 15/09/2009). Participants received aprepitant or placebo in a double-blind fashion during two 7-day treatment periods performed in a random order at 2-week interval. The experimental protocol was in accordance with the Helsinki Declaration and approved by the Institutional Review Board of the University Hospital of Rouen and the regional ethics committee (Comité de Protection des Personnes de Haute-Normandie). All participants provided written informed consent before participation.

*Participants.* All participants were recruited at the Centre for Clinical Investigation of the University Hospital of Rouen (CIC INSERM 1404) from 19 June 2009 to 19 March 2010. Eligible participants were adult male aged 18–30 years. Non-inclusion criteria were signs or history of liver, kidney, cardiovascular (including hypertension and orthostatic hypotension) and neurological dysfunction, and BMI > 27 kg m$^{-2}$. Mean body mass index was 22.4 ± 1.7 kg m$^{-2}$. None of them had taken any medications prior to the study. Clinical examination, including measurements of blood pressure and heart rate, blood count and plasma electrolyte levels, were normal (Supplementary Table 2).

*Interventions.* Aprepitant (125 mg at Day (D) 1 and 80 mg for D2–D7) and placebo were given orally at 08:00 a.m. to all volunteers by the investigators in a double-blind fashion during two 7-day treatment periods performed in a random order at 2-week interval (Supplementary Fig. 6). The aprepitant and placebo capsules were of identical appearance. All subjects maintained their regular diet during the study as confirmed by monitoring of natriuresis which remained globally stable. Such protocol of administration allows obtaining from D3 a plasma aprepitant concentration of 2.5–3 μM[41], a level much higher than the IC$_{50}$ of the molecule to the recombinant human NK1R (i.e. 0.1 nM)[12]. During the two treatment periods, the subjects underwent a series of pharmacological and physiological tests, including upright and supine positions at D4, a metoclopramide stimulation test at D5 and an insulin tolerance test at D7 (Supplementary Methods). Blood samples were obtained at 8:00 a.m. after 2 h of upright position at D4 and after 2 h of recumbency at D4 and D5 (10:00 a.m.). No side effect has been observed.

*Outcomes.* The effect of aprepitant on basal and stimulated corticosteroid secretion was appreciated by evaluating the variations of plasma aldosterone levels during the posture test (primary endpoint) and cortisol, renin, ACTH and electrolyte concentrations (secondary endpoints) in comparison with placebo regimen. The determination of plasma cortisol levels also allowed calculation of the aldosterone/cortisol ratio, which facilitates the interpretation of plasma aldosterone fluctuations by excluding the variations of mineralocorticoid secretion related to the nyc-themeral rhythm of ACTH. Assessment of basal production of aldosterone and cortisol has also been carried out by measurement of 24-h urinary aldosterone and free cortisol (secondary endpoint) from D3 to D4, before the beginning of stimulation tests. The impact of aprepitant treatment on hydromineral homeostasis and cardiovascular status has been evaluated by 24 h ambulatory blood pressure monitoring (secondary endpoint).

*Hormone assays.* Plasma and urine aldosterone, plasma ACTH, cortisol and renin levels were measured by immunoassays using respectively the following commercial kits: Immunotech RIA Aldosterone (Marseille, France), Immulite 2500 ACTH and cortisol (Siemens, Villepinte, France), Cisbio Renin III Generation (Saclay, France). Immunoassays used the automated workstation Synchron LXi 725 (Beckman Coulter, Villepinte, France). For determination of urine aldosterone and cortisol levels, immunoassays were preceded by a step of acid hydrolysis and steroid extraction by dichloromethane, respectively. Intra- and inter-assay variations were <10% for ACTH, cortisol, aldosterone and renin.

*Sample size.* Owing to the type of the trial (proof-of-concept study) and the absence of available previous clinical data on the influence of substance P on aldosterone production, it appeared that no statistical method could allow predetermining the number of subjects to be included in the study. Since a previous trial using a pharmacological antagonist targeting another paracrine adrenal regulatory factor (i.e. serotonin) showed that significant variations of plasma aldosterone levels could be observed in a sample of 18 healthy volunteers[42], we have decided to include 20 subjects.

*Randomisation.* The list of allocation sequence has been generated by the biostatistician (JF Ménard) using a random-numbers table. A Permuted Block Randomisation has been performed with four participants in each block. Centralised randomisation was performed with sequentially numbered containers. The list or

randomisation as well as the envelopes of blind lifting have been carried out by the methodology unit of Rouen University Hospital (JF Ménard). Briefly, the list of randomisation has been established by numbered envelopes, attributed in the chronological order of strict inclusion of volunteers. The treatment code (the nature of the treatment) assigned to each volunteer has been kept in sealed envelopes. Each volunteer has been randomised to either Group A or Group B, each group corresponding to the order of treatment assignment. The randomisation list has been then sent to the pharmacy of Rouen University Hospital (Dr N Donnadieu), which has conditioned the therapeutic units in boxes.

**Statistical analysis**. Data were collected in Microsoft Office Excel datasheets and expressed as median ± InterQuartile Range (IQR), minima and maxima or mean ± SEM. Relative changes were described as percentages of basal level. Data were analysed using the Prism program (GraphPad Software, Inc.). Statistical significance was assessed by non-parametric two-tailed Mann–Whitney test or Kruskal–Wallis test and Dunn's post-test after one-way ANOVA or two-way ANOVA for data obtained in vitro. For the clinical trial, biological parameters were analysed with the NCSS (Hintze J.; Kaysville, UT, USA) and StatXact-8 Crossover (Cytel Inc. Software,Antony, France) softwares to evaluate the period, treatment and carry-over effects by using the non-parametric Wilcoxon/mid rank matched pairs test. Statistical significance between kinetics and dose-response curves was analysed by Two-way ANOVA. $P$ values < 0.05 were considered significant.

**Reporting summary**. Further information on research design is available in the Nature Research Reporting Summary linked to this article.

## Data availability

The source data underlying Figs. 1a, 2a, 3a–d, 4a, b, 5b–f, and Supplementary Fig. 5 are provided as a Source Data file and available via the following link code https://zenodo.org/record/3736387#.XoSXUIgzaUk (https://doi.org/10.5281/zenodo.3736386). The data that support the findings of the clinical study are available from the corresponding author (H.L.) on reasonable request. These data are not publicly available due to them containing information that could compromise research participant privacy/consent. The human adrenal samples were obtained from the Tumor BioBank-Biological Resource Centre of the Rouen University Hospital under specific material transfer agreement and will not be available through the corresponding author.

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

## Acknowledgements

In vitro experiments were supported by the Institut National de la Santé et de la Recherche Médicale, the Conseil Régional de Normandie and the European Regional Development Fund. The clinical APHOS trial was an academic study which was sponsored by the principal investigator's host institution (HL; Rouen University Hospital). The trial was funded by the Rouen University Hospital and MSD Laboratories which also kindly provided aprepitant tablets. MSD had no priority access to the data which remained the property of the authors and their institution. No MSD employee contributed to the study. We thank Dr. Edgar-Yves Menguy and the staff at the Unité de Coordination des Prélèvements et Greffes (Rouen University Hospital) for their help with organ donation. We are grateful to Pr Jacques Weber (Centre d'Investigations Cliniques 1404), the Tumor BioBank-Biological Resource Centre of Rouen University Hospital directed by Pr Jean-Christophe Sabourin, the COMETE network, Dr Nathalie Donnadieu (Department of Pharmacy, Rouen University Hospital) and Julien Blot (Direction de la Recherche Clinique et de l'Innovation, Rouen University Hospital) for their collaboration, and Dr Celzo Gomez-Sanchez for his generous gift of CYP11B2 antibodies. We thank Carole Burel, Elodie Colas, Huguette Lemonnier and Valentina Pautasso for their skilful technical assistance. We also thank Pr P Lerouge for fruitful discussions. We are grateful to Dr Nancy Brown (Vanderbilt University Medical Center, Nashville, TN), Celzo Gomez-Sanchez (University of Mississippi Medical Center, Jackson, MS) and Constantine A Stratakis (National Institutes of Health, Bethesda, MD) for their helpful comments on our manuscript.

## Author contributions

H.L., E.L. and A.-F.C. conceived and designed research; J.W., C.D., A.-G.L., I.B., H.-G.B., C.D., S.C., F.G. and E.L. performed the experiments; G.D. facilitated sample collection; A.-F.C., C.G. and B.C. collected clinical and biological data for the subjects; J.W., C.D., J.F.M., H.L. and E.L. analysed data; J.W., E.L. and H.L. wrote the paper.

## Competing interests

H.L. received payment from MSD Laboratories for lecture unrelated to this study. J.W., C.D., A.-G.L., I.B., H.-G.B., C.D., S.C., F.G. and E.L declare no competing interests.
