## [Peer Review File · Nature Communications]

Reviewers' Comments:

Reviewer #1:

Remarks to the Author:

This study examines the role of SP and the NK1R in the regulation of aldosterone production in the adrenal gland. The authors examined the expression and localization of SP and its receptors in the adrenal gland by q-PCR and immune-staining, studied the effects of SP and NK1R antagonists on signaling, expression of synthetic enzymes, and hormone secretion in isolated adrenal gland cells. Finally, they completed a clinical trial that examined the effects of Aprepitant, an NK1R antagonist used clinically to treat nausea and vomiting, on secretion of steroid hormones in normal subjects. The study provides evidence for a role of SP and the NK1R in aldosterone secretion in humans.

The strength of this study is the novelty of the finding that SP and the NK1R control aldosterone secretion in human subjects. The results suggest that NK1R antagonists could be used to treat conditions associated with elevations in aldosterone secretion.

There are several issues that require attention.

Fig. 2. It is notoriously difficult to specifically detect GPCRs using antibodies. Problems include lack of antibody selectivity and challenges in immunoblotting for hydrophobic transmembrane proteins. While there are many reports using antibodies to the C-terminus of the NK1R, antibodies to the loop regions are less well characterized. The authors need to provide evidence of selectivity of both NK1R antibodies in the immunofluorescence and immunoblotting studies. Omission of the primary antibody is an inadequate control. They should examine whether the signals abolished by reabsorption of antibodies with the receptor fragments used for immunization. They need to provide evidence that the high and low MW bands correspond to full length and truncated NK1R. The authors should show complete gels to assess interaction of antibodies with other proteins. The two bands are of far higher MW than the full length and truncated NK1R – are they glycosylated? Please provide details of the peptides used for immunization to generate these antibodies.

Fig. 4. The analysis of signal transduction is rather superficial and some of the results are difficult to interpret. For the Ca signaling assays, does each line represent a single trace from one cell or a population of cells? It is impossible to tell which line of the SP trace corresponds to which concentration of SP as only 2 lines are labeled. The ERK data test only a single very high concentration of SP at one time point (60 min). It would be preferable to examine a time course response because ERK activation is rapid and transient. They should also test graded SP concentrations. Also, blots for pERK alone are shown; ERK should also be blotted as a control for loading and data normalized to total ERK. The authors should use NK1R antagonists to show that the Ca and ERK responses are mediated by the NK1R and not the NK2R, which will be activated by the very high concentrations of SP.

The authors conclude that the short form of the NK1R mediates the effects of SP on aldosterone secretion. There is no direct evidence to support this conclusion because the full-length receptor is expressed in the adrenal gland and there are no selective agonists or antagonists of the short form receptor. The pharmacological and signaling assays are too superficial to support this strong conclusion.

Reviewer #2:

Remarks to the Author:

In this study, J. Wils et al. have investigated the role of the neuropeptide "substance P" in the human adrenal. They show that substance P is expressed in the adrenal cortex, where it is mainly observed in nerve fibers crossing the cortex or under the capsule. In addition, they show that two

different isoforms of the tachykinin receptor NK1R are expressed in the adrenal.

The authors have used primary cells isolated from human adrenals to look at the role of substance P on the aldosterone production in vitro. Here, they show that especially the aldosterone production is enhanced when substance P is added and can be inhibited using an NK1R antagonist. This is mediated via the ERK pathway.

To investigate the role of substance P in vivo the authors have performed a double blind clinical study where healthy individuals received the NK1R antagonist aprepitant. As in the in vitro study the antagonist inhibited aldosterone production, however only in a supine position.

Globally, the manuscript is clear and well written. The experimental approaches used are appropriate and complementary providing a convincing description of the role of substance P in the human adrenal cortex. The manuscript brings substantial new information to previous articles on this topic.

In addition, the results are of significant importance as the described mechanism for aldosterone regulation independent of RAS might be a possible explanation for "aldosterone breakthrough", which can potentially be treated with NK1R antagonists.

Thus, the presented data will be of great interest for a large readership.

The authors should address the following issues:

Major comments:

1. The observation that substance P is released from nerves under the adrenal capsule and in the zG and zF is very interesting in terms of functional implications. However, why do the authors use S100 as a neural marker? S100 is more known as a marker of glia and Schwann cells. Why didn't the authors use for example beta III tubulin instead?

In relation to this, I think there is a mistake in the catalogue number for the S100 antibody. I wanted to check, which S100 protein your antibody recognizes, and by Dako I only found an S100 antibody with the number Z0311 instead of 20311 as in your table. The Z0311 antibody mainly recognizes S100B, which has also previously been observed to be expressed in both the murine adrenal cortex and medulla.

2. How do the authors explain that the SP-positive fibres are distinct from the adrenergic and cholinergic fibres?

3. It would be nice with a small description of the isolated cells (lines 123-124). Now, when you read the manuscript it is not clear if these are primary cells or a cell line. This is of course mentioned in the Materials and Methods, but this first comes later.

4. How do the authors explain that in vitro cortisol is slightly increased after the addition of substance P as they at the same time have shown that the NK1 receptor is barely expressed in the zona fasciculata? On the other hand, in the in vivo clinical study cortisol is increased when the NK1R is inhibited, which the authors explain with a stimulation of the HPA axis?

5. A small model showing how substance P could potentially stimulate aldosterone production would be nice.

Minor comments

1. Line 80: "a NK1R" has to be replaced with "an NK1R"

2. Line 106: I would change "recognizes therefore" to "therefore recognizes"

3. Line 181: "mineralo..." instead of "mineraolo..."

4. Line 251: "Total RNAwere extracted by using Tri Reagent" has to be changed to "Total RNA.....was extracted using Tri Reagent"

5. Line 254. "ARN" has to be changed to "RNA"

6. Line 271. "with" is missing in "treated peroxidase"

7. Figure 1, D;E;F: In B and C the different zones of the adrenal are marked. This information is missing in D-F.

8. Figure 2A. In panel A there is a mistake in the "unit", which is now mentioned as "10⁻²"

9. Figure 2B. In the upper part it would be nice with a legend next to the bands of the WB (NK1R and NK1Rs).

10. Figure 2E. The size of the scale bar is missing

11. In the legend to Fig. 2C,D "NK1 receptor" should be removed as the abbreviation has already

been used several times before.

12. Figure 3C. Here it would be nice with a legend in the figure as in Figure 3A and B showing which curve is what.

13. In figure 5 many of the lines are oblique. They are not really parallel to the x or y axis.

14. In figure 5 and supplementary figure 7 it is not consistent if aldosterone and cortisol are written with capital letters or not.

Reviewer #1:

This study examines the role of SP and the NK1R in the regulation of aldosterone production in the adrenal gland. The authors examined the expression and localization of SP and its receptors in the adrenal gland by q-PCR and immune-staining, studied the effects of SP and NK1R antagonists on signaling, expression of synthetic enzymes, and hormone secretion in isolated adrenal gland cells. Finally, they completed a clinical trial that examined the effects of Aprepitant, an NK1R antagonist used clinically to treat nausea and vomiting, on secretion of steroid hormones in normal subjects. The study provides evidence for a role of SP and the NK1R in aldosterone secretion in humans.

*- The strength of this study is the novelty of the finding that SP and the NK1R control aldosterone secretion in human subjects. The results suggest that NK1R antagonists could be used to treat conditions associated with elevations in aldosterone secretion.
- There are several issues that require attention.*

We thank the reviewer for his highly accurate and relevant comments which were very helpful to resolve the weaknesses of our study.

Fig. 2. It is notoriously difficult to specifically detect GPCRs using antibodies. Problems include lack of antibody selectivity and challenges in immunoblotting for hydrophobic transmembrane proteins. While there are many reports using antibodies to the C-terminus of the NK1R, antibodies to the loop regions are less well characterized. The authors need to provide evidence of selectivity of both NK1R antibodies in the immunofluorescence and immunoblotting studies. Omission of the primary antibody is an inadequate control. They should examine whether the signals abolished by reabsorption of antibodies with the receptor fragments used for immunization. They need to provide evidence that the high and low MW bands correspond to full length and truncated NK1R. The authors should show complete gels to assess interaction of antibodies with other proteins. The two bands are of far higher MW than the full length and truncated NK1R – are they glycosylated? Please provide details of the peptides used for immunization to generate these antibodies.

General considerations.

We thank the reviewer for addressing this important issue. We fully agree that immunological characterization of membrane receptors suffers from several limitations inherent to this approach, the first of them being the only partial specificity of the antibodies used for IHC/IHF studies. Negative controls are thus important to assess the specificity of the antibodies in the tissue of interest. We are aware that omission of the primary antibody is an imperfect control as it only allows investigating nonspecific binding of the secondary antibody. The “absorption control” has in fact been proposed to support the specificity of immunolabeling of a molecule. However, it is now considered as a rather weak control since it only shows that the antibody is specific for the antigen to which it was prepared but cannot demonstrate that the antibody is fully specific for the targeted molecule in the tissue (please, see Holmseth et al., J Histochem Cytochem, 2012, 60:174-87; see also the standards of the Histochemical Society published by Hewitt et al., J Histochem Cytochem, 2014, 62:693-7). Especially, the failure of pre-absorption to abolish the signal can sometimes result from the fact that the conformation of the sought molecule in the tissue may be different from that of the immunogenic free

peptide in the incubation buffer (please, see: True, *Histochem Cell Biol*, 2008, 130:473-480). In addition, in many studies including ours, this type of experiment is not feasible since the manufacturers do not commercialize the immunogenic peptides that have been used to generate their antibodies and even sometimes refuse to communicate the primary structures of the latter. Therefore, it appears now that characterization of receptors cannot be exclusively based on immunological techniques. In the present work, the demonstration that the NK1R is involved in the regulation of aldosterone secretion by substance P is based on several complementary approaches. In particular, the largely predominant localization of NK1R immunoreactivity in the subcapsular region of the adrenal cortex (especially, the intense labeling in aldosterone-producing cells) was fully consistent with the capacity of substance P to preferentially activate aldosterone production. The involvement of the NK1R in the mineralocorticoid response to SP is further supported by the inhibitory action of the NK1R antagonist aprepitant on SP-evoked aldosterone secretion *in vitro* as well as the repressive effect of aprepitant on basal aldosterone production *in vivo*. Activation of the calcium and ERK signaling pathways was also consistent with what is known about the transduction mechanisms associated with the NK1R (Lai JP, et al *Proc. Natl. Acad. Sci. U. S. A.*, 2008, 105:12605–12610). Lastly, we can also mention that these results were somewhat expected as the NK1R mediates the action of SP in numerous tissues (please, see Schank and Heilig, *Int Rev Neurobiol*, 2017, 136:151-175).

Specific considerations

Regarding the antibodies which can be used for IHC detection of the NK1R, we agree with the reviewer that antibodies directed against the C-terminus of the receptor have been more widely used than antibodies to the transmembrane loops. However, the use of diverse antibodies against different epitopes is interesting as they can provide complementary data. In fact, antibodies to the intracellular tail of the receptor should only detect the long isoform of the receptor while antibodies to loop regions theoretically allow detection of both long and short isoforms. In addition, antibodies against the transmembrane region of the receptor, like the SAB4502913 antibody (Sigma-Aldrich; an affinity-purified rabbit polyclonal antibody developed against the region 211-260 amino acids of the human NK1R) have been successfully used to detect expression of the NK1R in the central nervous system (Peirs C et al., *Neuron*, 2015;87:797-812; Yackle K et al., *Science*, 2017, 355:1411-1415). We have thus purchased this antibody and performed additional IHC and immunoblotting studies in addition to the experiments previously shown in the initial version of the manuscript. To summarize, three distinct commercial antibodies to the NK1R receptor (the available information about the immunogenic peptides used for their generation is now mentioned in Supplemental Table 4) have been employed for adrenal western blot and immunohistological studies and gave the following results:

- T5950 (Sigma-Aldrich), an affinity-purified rabbit polyclonal antibody developed against the 2nd extracellular loop of human NK1R (García-Ortega, J. et al. *Biol. Reprod.*, 2016, 94:124), showed a highly predominant 70 kDa band at western blotting (WB).
- WB using SAB4502913 antibody showed bands of 70 and 35 kDa.
- PA3-301 (ThermoFisher Scientific), a rabbit polyclonal antibody developed against the C-terminal region between the 387 and 407 amino acids of the human NK1R long isoform, has already been used to localize the NK1R in human spinal cord tissues (Leonard et al., *Spinal Cord*, 2014, 52:17-23). By use of this antibody, we observed the presence of 115, 70 and 55 kDa bands at WB.

- In addition, the 3 antibodies allowed detection of NK1R immunoreactivity in the ZG and, more weakly, in the ZF. NK1R immunolabeling was also observed in adrenal nervous ganglia and artery walls which can be considered as positive control tissues. The latter is now presented in Suppl Fig. 2.

The 35 and 55 kDa visualized at WB are known to correspond to the short and long variants of the NK1R. The 70 and 115 kDa bands have already been reported by other groups in various tissues and may correspond to the ubiquitinated forms of the receptor (Cottrell, G. S. et al. J. Biol. Chem., 2006, 281:27773–27783). Their presence may also result from other post-translational modifications, like glycosylation and phosphorylation, which are known to affect the NK1R (Tansky, M. F., et al. Proc. Natl. Acad. Sci. U. S. A., 2007, 104:10691–10696). Regarding post-translational maturation of NK1R, we have performed some preliminary deglycosylation studies which suggest that the 70 kDa band corresponds to a glycosylated form of the protein (data not shown).

Globally, these data indicate that the two isoforms of the receptor, whose mRNA are detected in the adrenal extracts, are expressed in the human adrenal but do not allow determining which variant predominantly mediates the action of substance P on aldosterone secretion.

Fig. 4. The analysis of signal transduction is rather superficial and some of the results are difficult to interpret. For the Ca signaling assays, does each line represent a single trace from one cell or a population of cells? It is impossible to tell which line of the SP trace corresponds to which concentration of SP as only 2 lines are labeled. The ERK data test only a single very high concentration of SP at one time point (60 min). It would be preferable to examine a time course response because ERK activation is rapid and transient. They should also test graded SP concentrations. Also, blots for pERK alone are shown; ERK should also be blotted as a control for loading and data normalized to total ERK. The authors should use a NK1R antagonists to show that the Ca and ERK responses are mediated by the NK1R and not the NK2R, which will be activated by the very high concentrations of SP.

We agree with the reviewer that the analysis of the transduction mechanisms associated with the adrenal NK1R needed clarification.

Regarding the impact of substance P on the calcium intracellular pathway, we have now clearly mentioned that calcium signalling is illustrated as mean of fluorescence intensities simultaneously recorded in at least three distinct wells (each containing 2×10^5 cells) (please, see page 11, lines 304-305, 311-313). In addition, the kinetics profiles of the cytosolic calcium concentration in response to the increasing doses of the peptide are now presented with distinct colours to improve the clarity of Figure 4. We have also added a dose-response (SP concentration/fluorescence peak value) curve together with histograms presenting the effect of ATP and Ang II, in order to plainly show that the increase in cytosolic calcium concentration induced by substance P is very low in comparison with that triggered by ATP and Ang II. These findings indicate that the calcium signalling pathway is marginally involved in the transduction signal elicited by substance P consistently with the fact that substance P has no significant influence on *CYP11B2* expression which is known to be strongly activated by intracellular calcium (Hattangady *et al.*, Mol Cell Endocrinol, 2012, 24, 350:151-62). We thus did not feel necessary to investigate the calcium response to substance P in the presence of aprepitant.

We agree with the reviewer that evaluation of the action of substance P on the ERK pathway necessitated additional experiments to obtain a better picture of activation of the process in

adrenocortical cells. In order to satisfy the reviewer's recommendation, we have thus investigated the influence of increasing concentrations of substance P on ERK phosphorylation as well as the time course of the pERK response. We have also normalized pERK data to ERK as requested. As hypothesized by the reviewer, the pERK response was found to be rapid (5 min) and transient (with a decrease towards basal levels at 15 min). ERK phosphorylation induced by substance P was also dose-dependent, appearing at 10^{-9} M and reaching a maximum at 10^{-7} M (EC50 = 1.1 ± 01 nM). These data are consistent with the known affinity of substance P for the NK1R (nanomolar) and argue against a significant contribution of the NK2R for which substance P has a weak affinity (around 10^{-7} M; please see Page *et al.*, Proc Natl Acad Sci USA, 2003, 100:6245-50). A possible contribution of the NK2R to the adrenal response to SP at high concentration (10^{-6} M) can be excluded on the basis that the NK2R antagonist GR has no influence on SP-evoked aldosterone secretion by cultured adrenocortical cells at this concentration (please, see Fig. 3D). This finding is also coherent with the observation that *TACR2* mRNA levels are barely detectable in adrenal extracts (*i.e.* 100 times lower than *TACR1* mRNA rates; please mind the gap in the y-axis scale of the graph presented in Fig. 2A). We can also assume that the amount of functional NK2R in adrenocortical tissues is certainly very low since NKA, which is known to have a strong affinity for the NK2R (nanomolar; please, see Rupniak *et al.*, Plos One, 2018, 13:e0205894), only stimulates aldosterone production at high concentrations (≥ 100 nM). Considering these findings and owing to the fact that normal human adrenal glands are not easily available, we have chosen to focus our additional experiments on the time- and dose-response curves of pERK to SP rather than evaluating the impact of NK1R and NK2R antagonists on SP-evoked activation of the ERK pathway.

The authors conclude that the short form of the NK1R mediates the effects of SP on aldosterone secretion. There is no direct evidence to support this conclusion because the full-length receptor is expressed in the adrenal gland and there are no selective agonists or antagonists of the shorty form receptor. The pharmacological and signaling assays are too superficial to support this strong conclusion.

We agree with the reviewer that our conclusion sentence claiming that substance P stimulates aldosterone production through the truncated form of the NK1R is an overstatement. We have thus suppressed any reference to NK1R isoforms in this sentence which now only indicates that substance P activates aldosterone release *via* the NK1R.

Reviewer #2 (Remarks to the Author):

In this study, J. Wils et al. have investigated the role of the neuropeptide "substance P" in the human adrenal. They show that substance P is expressed in the adrenal cortex, where it is mainly observed in nerve fibers crossing the cortex or under the capsule. In addition, they show that two different isoforms of the tachykinin receptor NK1R are expressed in the adrenal.

The authors have used primary cells isolated from human adrenals to look at the role of substance P on the aldosterone production in vitro. Here, they show that especially the aldosterone production is enhanced when substance P is added and can be inhibited using an NK1R antagonist. This is mediated via the ERK pathway. To investigate the role of substance P in vivo the authors have performed a double blind clinical study where healthy individuals received the NK1R antagonist

apreitant. As in the in vitro study the antagonist inhibited aldosterone production, however only in a supine position.

Globally, the manuscript is clear and well written. The experimental approaches used are appropriate and complementary providing a convincing description of the role of substance P in the human adrenal cortex. The manuscript brings substantial new information to previous articles on this topic.

In addition, the results are of significant importance as the described mechanism for aldosterone regulation independent of RAS might be a possible explanation for “aldosterone breakthrough”, which can potentially be treated with NK1R antagonists. Thus, the presented data will be of great interest for a large readership.

We thank the reviewer for his very positive appreciation and valuable comments which were very useful for the improvement of our manuscript.

The authors should address the following issues:

Major comments:

1. The observation that substance P is released from nerves under the adrenal capsule and in the zG and zF is very interesting in terms of functional implications. However, why do the authors use S100 as a neural marker? S100 is more known as a marker of glia and Schwann cells. Why didn't the authors use for example beta III tubulin instead? In relation to this, I think there is a mistake in the catalogue number for the S100 antibody. I wanted to check, which S100 protein your antibody recognizes, and by Dako I only found an S100 antibody with the number Z0311 instead of 20311 as in your table. The Z0311 antibody mainly recognizes S100B, which has also previously been observed to be expressed in both the murine adrenal cortex and medulla.

In fact, protein S100 has been widely used as a neural marker to describe innervation of various organs (please, see for example the recent paper by Lee et al., Cancer Genomics Proteomics, 2018, 15:337-342). However, we agree with the reviewer that it is rather a glia and Schwann cell marker. In order to satisfy the reviewer's comment, we have thus performed additional immunohistochemical studies with antibodies to a validated pan-neuron marker. We have chosen antibodies to PGP9.5 (a protein of the cytoskeleton) which have previously been used to describe the global innervation of the human adrenal gland (McNicol et al., Acta Anat (Basel), 1994, 151:120-3). These new results are shown in Fig. 1D where they replace the protein S100 data which have been suppressed. As expected, they are found to be consistent with the known distribution of adrenal nerve fibres, a subpopulation of them being labelled by substance P antibodies.

For the reviewer's information, we confirm that the correct reference for the Dako PS100 antibody was Z0311 and not 20311. We apologize for this typing mistake.

2. How do the authors explain that the SP-positive fibres are distinct from the adrenergic and cholinergic fibres?

In many organs, substance P fibres are distinct from adrenergic and cholinergic fibers and are thus representative of the non-adrenergic non-cholinergic (NANC) system which is thought to play an important physiological role in the gastrointestinal tract and airways (please, see for review: Maggi, Regul Pept, 2000, 93:53-64 & Kraneveld and Nijkamp F, Int Immunopharmacol, 2001, 1:1629-50). The NANC system is considered as the third

component of the autonomic nervous system by some authors. This precision has been added on page 4, lines 86-88. It is interesting to highlight that the observation of the presence of substance P-positive fibres in the same nerve trunks as sympathetic fibres is not restricted to the adrenal gland but has also been reported in other organs like the human pancreas (Chien HJ et al Am J Physiol Gastrointest Liver Physiol., 2019, doi: 10.1152/ajpgi.00116.2019).

3. It would be nice with a small description of the isolated cells (lines 123-124). Now, when you read the manuscript it is not clear if these are primary cells or a cell line. This is of course mentioned in the Materials and Methods, but this first comes later.

As suggested by the reviewer, it is now clearly mentioned that isolated cells are adrenocortical cells in primary culture on page 5, lines 124-125.

4. How do the authors explain that in vitro cortisol is slightly increased after the addition of substance P as they at the same time have shown that the NK1 receptor is barely expressed in the zona fasciculata? On the other hand, in the in vivo clinical study cortisol is increased when the NK1R is inhibited, which the authors explain with a stimulation of the HPA axis?

In fact, NK1R immunoreactivity is predominantly present in the ZG but is also detected less intensely in some cell cords of the ZF. This observation is thus consistent with the in vitro cortisol response to substance P which appears significant at only high concentrations of the neuropeptide (from 10^{-7} M). On the other hand, we agree with the reviewer that the increase in cortisol levels observed in healthy volunteers after administration of aprepitant (which remains to be confirmed as it was not significant in our study) cannot be the result of the impact of the drug on the adrenal glands since substance P activates cortisol synthesis *in vitro*. In agreement with this interpretation, we did not observe any decrease of plasma ACTH levels concomitantly with cortisol elevation in the tested volunteers. In fact, the plasma cortisol/ACTH ratio remained unchanged under aprepitant treatment suggesting therefore an impact of the drug on the corticotropic axis. These data are shown in supplemental materials (Fig S4). It is conceivable that substance P may exert complex actions at the hypothalamo-pituitary level involving both an inhibitory effect *via* the NK1 receptor and stimulatory effect through activation of other receptor types like the NK2 and NK3 receptors which are known to be expressed in hypothalamic neurons and pituitary cells (Nussdorfer & Malendowicz, Peptides, 1998, 19: 949-968). If this hypothesis holds true, aprepitant may reveal the stimulatory effect of substance P by antagonizing the inhibitory action of the peptide. Of course, this interpretation is highly speculative.

5. A small model showing how substance P could potentially stimulate aldosterone production would be nice.

As suggested by the reviewer, we have added a scheme presenting the mechanisms by which substance P could modulate aldosterone production (please, see Fig. 6).

Minor comments

1. Line 80: “a NK1R” has to be replaced with “an NK1R”

We apologize for this mistake which has been corrected in the revised version of the manuscript (please, see page 3, line 75).

2. Line 106: I would change “recognizes therefore” to “therefore recognizes”

The result section has been deeply modified to include the new data requested by reviewer 1. Consequently, the word “therefore” has been removed from the sentence (please, see page 4, lines 100-103)

3. Line 181: “*mineralo...*” instead of “*mineraolo...*”

This typing mistake has been corrected (please, see page 7, line 186).

4. Line 251:” *Total RNAwere extracted by using Tri Reagent” has to be changed to “Total RNA.....was extracted using Tri Reagent”*

This mistake has been corrected (please, see page 9, line 255).

5. Line 254. “*ARN*” has to be changed to “*RNA*”

This typing mistake has been corrected (please, see page 9, line 258).

6. Line 271. “*with*” is missing in “*treated peroxidase*”

We apologize for this omission which has been corrected (please, see page 10, line 277).

7. Figure 1, D;E;F: *In B and C the different zones of the adrenal are marked. This information is missing in D-F.*

As requested by the reviewer, the labels indicative of the different zones of the cortex have been added in the panels D-F of Figure 1.

8. Figure 2A. *In panel A there is a mistake in the “unit”, which is now mentioned as “10-2”*

As requested by the reviewer, the correct unit is now indicated in the y-axis label in the panel A of Figure 2.

9. Figure 2B. *In the upper part it would be nice with a legend next to the bands of the WB (NK1RI and NK1Rs).*

The panel has been deeply modified to include the new data requested by reviewer 1. It appears that interpretation of the western blot profiles is complex. The use of three distinct antibodies directed to diverse regions of the NK1R allows detection of several bands which correspond to the short and long variants as well as glycosylated, phosphorylated and ubiquitinated forms of the receptor. Please, see the results section on pages 4-5, lines 100-113.

10. Figure 2E. *The size of the scale bar is missing*

We apologize for this omission which has been corrected in the revised version of Figure 2.

11. *In the legend to Fig. 2C,D “NK1 receptor” should be removed as the abbreviation has already been used several times before.*

The mention “NK1 receptor” has been suppressed from the legend to Figure 2 in order to satisfy the reviewer’s recommendation.

12. Figure 3C. *Here it would be nice with a legend in the figure as in Figure 3A and B showing which curve is what.*

As proposed by the reviewer, a legend identifying each curve has been added in Figure 3C.

13. *In figure 5 many of the lines are oblique. They are not really parallel to the x or y axis.*

Figure 5 has been redrawn to obtain a perfect representation of the lines.

14. *In figure 5 and supplementary figure 7 it is not consistent if aldosterone and cortisol are written with capital letters or not.*

The terms “aldosterone” and “cortisol” are now written without capital letters in all cases to avoid any inconsistency between panels of the 2 figures.

Reviewers' Comments:

Reviewer #1:

Remarks to the Author:

Thank you for addressing my concerns. I have 2 suggestions for revision.

1. Please show the entire blot for Fig 2B as for 2C and D. This will help readers to understand the selectivity of the three different NK1R antibodies.

2. I appreciate the difficulty in obtaining human adrenal cells. However, if available please show whether aprepitant inhibits SP-evoked ERK signaling. Given that you implicate ERK in hormone release, and show that aprepitant blocks hormone release, these data would be a useful addition.

Reviewer #2:

Remarks to the Author:

the authors have significantly improved their manuscript. I have no further concerns or Statements.

Responses to Reviewer 1

1. *Please show the entire blot for Fig 2B as for 2C and D. This will help readers to understand the selectivity of the three different NK1R antibodies.*

The entire blots, whose parts are presented in Fig. 2, are now provided in Supplementary Fig. 2.

2. *I appreciate the difficulty in obtaining human adrenal cells. However, if available please show whether aprepitant inhibits SP-evoked ERK signaling. Given that you implicate ERK in hormone release, and show that aprepitant blocks hormone release, these data would be a useful addition.*

We agree with the reviewer that showing an inhibition of SP-induced ERK activation by aprepitant would be a relevant additional piece of information. However, we believe that it can be reasonably concluded that aldosterone stimulates aldosterone secretion through activation of the NK1 receptor and the ERK pathway by taking together the already available results. These observations are the following: (i) the stimulatory action of substance P (SP) on aldosterone production is blunted by ERK inhibitor; (ii) the effect of SP on aldosterone is inhibited by aprepitant and, (iii) SP activates ERK phosphorylation. In addition, as highlighted by the reviewer, normal human adrenal tissue is quite rare and investigating the impact of aprepitant on SP-evoked ERK signalling could necessitate several months. We have thus chosen to address these issues in the Discussion section (on page 8, lines 195-200) rather than perform additional experiments.